# Persistent red blood cells retain their ability to move in microcapillaries under high levels of oxidative stress

Nadezhda A. Besedina [1,5], Elisaveta A. Skverchinskaya[2,5], Stanislav V. Shmakov[1], Alexander S. Ivanov[3], Igor V. Mindukshev[2] & Anton S. Bukatin [1,4 ✉]

Oxidative stress is one of the key factors that leads to red blood cells (RBCs) aging, and impairs their biomechanics and oxygen delivery. It occurs during numerous pathological processes and causes anaemia, one of the most frequent side effects of cancer chemotherapy. Here, we used microfluidics to simulate the microcirculation of RBCs under oxidative stress induced by *tert*-Butyl hydroperoxide. Oxidative stress was expected to make RBCs more rigid, which would lead to decrease their transit velocity in microfluidic channels. However, single-cell tracking combined with cytological and AFM studies reveals cell heterogeneity, which increases with the level of oxidative stress. The data indicates that the built-in antioxidant defence system has a limit exceeding which haemoglobin oxidation, membrane, and cytoskeleton transformation occurs. It leads to cell swelling, increased stiffness and adhesion, resulting in a decrease in the transit velocity in microcapillaries. However, even at high levels of oxidative stress, there are persistent cells in the population with an undisturbed biophysical phenotype that retain the ability to move in microcapillaries. Developed microfluidic analysis can be used to determine RBCs' antioxidant capacity for the minimization of anaemia during cancer chemotherapy.

---

[1] Department of Physics, Alferov University, Saint-Petersburg, Russia. [2] Sechenov Institute of Evolutionary Physiology and Biochemistry of the RAS, Saint-Petersburg, Russia. [3] Peter the Great St.Petersburg Polytechnic University, Saint-Petersburg, Russia. [4] Institute for Analytical Instrumentation of the RAS, Saint-Petersburg, Russia. [5]These authors contributed equally: Nadezhda A. Besedina, Elisaveta A. Skverchinskaya. ✉email: antbuk.fiztek@gmail.com

Delivery of oxygen and nutrients to tissues, removal of $CO_2$, and metabolic products are provided by specialized blood cells—erythrocytes (red blood cells, RBCs). These functions are implemented when RBCs penetrate narrow capillaries, the diameter of which is smaller or comparable to the diameter of the RBCs themselves[1] due to their particular biconcave disc shape[2,3] and specific mechanical properties[4]. After changing the RBC's mechanical phenotype, their functional viability is impaired. They can no longer squeeze into the capillaries, where the main gas exchange occurs; therefore, such cells are removed from the circulation in the spleen[5]. Moreover, worsening RBC deformability can determine or accompany different pathologies such as sepsis[6], malaria[7,8], sickle cell anemia[9], diabetes[10], and hereditary disorders[11].

One of the most common causes of changing mechanical phenotype of red blood cells is oxidative stress. It is a multi-stage process, which primarily triggers damage to RBC's membrane and cytoskeleton proteins[12–15]. Eventually, it leads to red blood cells aging, impairs their biomechanics and oxygen delivery function[16]. One of the important cases of RBCs oxidative stress is the intake of xenobiotics, particularly cytostatics and other antitumor drugs[17,18], which leads to anemia, one of the most frequent side effects of cancer chemotherapy. Another critical case of RBCs oxidative stress is the storage of donated blood due to dysregulation of antioxidant pathways that limit its shelf life[19].

Nowadays, there are several single-cell experimental methods for quantitative evaluation of the mechanical properties of cells. They include micropipette aspiration[20], optical stretching[21,22], flickering analysis[23], atomic force microscopy (AFM)[24], and ultrasounds[25]. However, all these methods are not applicable in clinical practice, thanks to time-consuming and laborious procedures. RBCs functional disorders in patient-contact experience are most often assessed by ektacytometry[26] and osmotic fragility test[27]. The first method expresses RBCs deformability as an elongation index, calculated by laser diffraction patterns from the entire RBCs population versus applied shear stress. The second one determines deformability as the hemolysis resistance of the RBCs population. Both of them are indirect and do not estimate mechanical parameters of the RBCs under a microflow state. Besides, ektacytometry is insensitive to the appearance of minor populations of poorly deformed RBCs because it estimates the entire population's deformability but not of each cell separately[28].

Microfluidic technologies provide new opportunities for biophysical phenotyping of single cells under microcirculation-mimicking conditions[29,30]. Their main advantage is the high speed of statistics collection, which the above-mentioned single-cell techniques cannot grant. One of the most prominent microfluidic methods is constriction-based deformability cytometry[31–34]. Its essence lies in measuring the time of RBCs passage of the microfluidic channel constriction, which has a cross-section lower or comparable to RBCs diameter. Other methods for mechanical phenotyping are shear flow deformability cytometry[35] and extensional flow deformability cytometry[36], which are based on assessing the response of the single single-cell shape to the shear rate of the flow.

These techniques were successfully used to determine biomechanical deviations of transformed RBCs both under the action of xenobiotics[37,38] and various diseases[39–41]. Today, microfluidic devices make it possible to assess RBCs' deformability, aggregation, and hematocrit simultaneously[42]. Moreover, they can even mimic complex microcapillary networks of the lung[43] and the eye retina[44] for a detailed study of blood microcirculation, including occlusion emergence. Thus, many research groups have presently demonstrated the ability to detect certain RBC disorders using microfluidic devices. However, little is

known about the relationship between the mechanical properties of oxidatively stressed RBCs and their characteristics routinely measured in clinics.

In this work, we simulate the microcirculation of RBCs in a microfluidic device under various levels of oxidative stress induced by *tert*-Butyl hydroperoxide (tBuOOH). Oxidative stress was expected to make RBCs more rigid, which would lead to decrease their transit velocity in microfluidic channels. However, single-cell tracking reveals the splitting of the population into two subpopulations with different transit velocities at concentrations of tBuOOH 0.7 mM and higher. To identify the reason for such splitting, we made cytological analysis and AFM studies of the cells under oxidative stress. Our data show the decrease of esterase activity followed by swelling, stiffness, and adhesion increment caused by membrane transformation in "slow" cells. However, even at high concentrations of tBuOOH, there are persistent cells that move in the channels as fast as the untreated cells and retain their morphology and membrane structure. These findings open a new perspective on the oxidative stress damage of RBCs and are valuable for the estimation of cancer chemotherapy side effects.

## Results

**Microfluidic device for simulation of microcirculation**. Human RBCs are biconcave disks with a typical diameter of 6.2–8.2 μm and a maximum thickness of 2–2.5 μm[45]. This shape allows RBCs to deform and pass microcapillaries whose diameter can be less than 5 μm. To investigate how oxidative stress influences this ability, we developed a device (Fig. 1) with 16 microchannels with sizes comparable to RBCs dimensions (~2.5 μm wide and 8.0 μm high) to evaluate the velocity of the erythrocyte's passage (Supplementary Fig. 1). The microchannel's length was 200 μm for a reliable determination of the transit velocity[14], and the alignment chambers provided a more accurate entry of RBCs into microchannels[46]. The device was made from PDMS Sylgard 184 (Dow Corning) using soft lithography technique[47,48] using a reactive ion etched silicone mold.

RBCs were introduced into the microfluidic device under constant hydrostatic pressure at the concentration of $0.5 \times 10^8$ cells/mL to ensure that only one cell passed through a microchannel simultaneously. Cells' transit velocity was determined by analyzing videos obtained by a brightfield optical microscope at 400 frames/s. To receive statistically correct data, the number of measured RBCs during one experiment was 250–700 in control and 150–500 under the action of oxidative stress in 9–10 different channels in the same device. (Supplementary Video 1 and Video 2). The obtained values of the cell's velocity were in the range 6–10 mm/s which is comparable with in vivo velocities[49]. To compare data from different experiments, we normalized cells' velocities in microchannels to the average velocities of the fluid flow in these microchannels. Therefore, the velocities are presented in the form of arbitrary units.

**RBCs transport in microfluidic microcapillaries**. To investigate how oxidative stress influences RBCs' microcirculation, they were extracted from blood samples collected by venipuncture from 18 healthy volunteers, both sexes, the age median—32.5 (22–68). Oxidative stress in the cells was induced by tBuOOH in 0.1–1.5 mM concentrations. It is an organic oxidative compound widely used in oxidation processes as a selective and inexpensive oxidizing agent[12–15,50]. It causes rapid glutathione oxidation and reactive oxygen species (ROS) formation in reactions between tBuOOH and hemoglobin. The ROSs immediately react with polyunsaturated fatty acids in membranes, form reactive aldehydes and further deplete cellular stores of glutathione, leading to

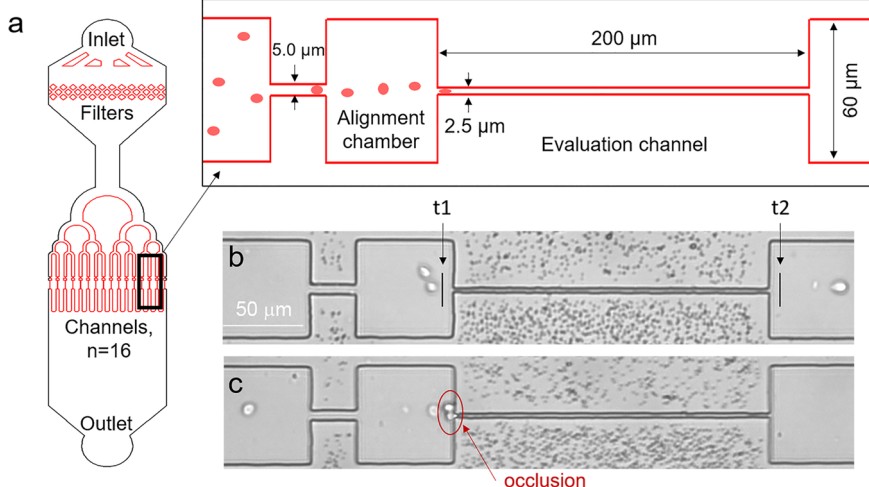

**Fig. 1 Microfluidic device for investigation of RBCs transport in microcapillaries. a** Schematic view of the device; **b** an image of a single microchannel with control RBCs; **c** 1.5 mM tBuOOH treatment can lead to microchannel's occlusion. t1 and t2 marks indicate the capturing points for determining cells' transit time in microchannels.

disruption of membrane structure. To acquire correct results in all the experiments, we kept the concentration of erythrocytes during incubation for 4–5 h at 37 °C constant at $0.5 \times 10^9$ cells/mL[15]. The oxidative stress model was validated by staining RBCs with the DCF-DA dye (35845, Sigma-Aldrich), which was sensitive to the presence of ROS by flow cytometry analysis (Supplementary Fig. 2).

Obtained results showed that RBCs subjected to oxidative stress changed the ability to pass microchannels. It was shown that the distribution of relative velocities of native RBCs could be approximated by a Gaussian function with $X_c = 0.309$ a.u. ($\sigma = 0,088$ a.u., $n = 18$ donors), which means that the cells moved approximately three times slower than the fluid flow in these channels (Fig. 2a). In the samples treated with 0.1, 0.3, and 0.5 mM of tBuOOH, the Gaussian distribution of relative velocities retained values $X_c = 0.311$ a.u. ($\sigma = 0.076$ a.u., $n = 12$ donors), $X_c = 0.316$ a.u. ($\sigma = 0.093$ a.u., $n = 12$ donors), and $X_c = 0.310$ a.u. ($\sigma = 0.075$ a.u., $n = 12$ donors), respectively (Fig. 2a).

After incubating RBCs in the presence of 0.7 mM tBuOOH, a pool of cells with significantly lower velocities (less than 0.1675 a.u.), which we called "slow RBCs" in Fig. 2, appeared ($n = 12$ donors). Therefore the distribution of cells' velocities became bimodal with the second mode at the range of velocities less than 0.1675 a.u. The threshold 0.1675 a.u. was chosen as the intersection point between two modes of the distribution (Fig. 2b). While the second mode of the distribution appeared in the region of low velocities, the main mode did not change its position ($X_c = 0.316$ a.u., $\sigma = 0.089$ a.u.).

With a further increase in the oxidizer concentration, the number of cells with normal velocities decreased, whereas the percentage of the slow cells grew up to 51.1% at 1.5 mM tBuOOH (Fig. 2c, d, $n = 5$ donors). Besides, we registered massive occlusions at tBuOOH concentrations of 0.7 mM and higher. To be more detailed, we determined an occlusion as an event in which the RBC stands in front of the channel entrance for ten or more frames (~22 ms). The occlusion cases ratio was up to ~76% at 1.5 mM of tBuOOH (Supplementary Fig. 3).

**RBCs sizes and population heterogeneity.** One of the reasons for the impaired passage of the RBCs through microchannels caused by oxidative stress could be a change in their volume and thickness. It might happen due to violations in cytoskeleton and

membrane organization[51]. According to the hematological analysis ($n = 12$ donors) made by Medonic-M20 (Boule Medical A.B., Sweden), exposure to tBuOOH resulted in cell swelling (Fig. 3a and b). At tBuOOH concentrations of 0.5 mM and 0.7 mM, the increase of mean corpuscular volume (MCV) was statistically significant ($p = 0.0023$ and $p = 0.00007$, respectively) but did not exceed the physiological range (70–95 fL). However, when tBuOOH concentration was 1.5 mM, a meaningful increase in cells' volume was recorded: the MCV was 111.4 fL (95.4–124.3) median, which was 1.32 times higher than the MCV of untreated cells (Fig. 3b).

Moreover, our data show that oxidative stress led to an increase in the RBCs population heterogeneity, which can be expressed by the red blood cell distribution width (RDW-SD) of complete blood count (Fig. 3c). The determination of RDW-SD[52] is a direct measurement of the erythrocyte volume histogram width at 20% of the curve height. It is measured in fL (femtolitres) and reflects the difference between the maximum and minimum RBCs cell volume in the test sample. Thus, RDW-SD is a more sensitive index than routinely used RDW% (RBC volume histogram width at 50% of the curve height to MCV ratio) in cases when a small number of macrocytes and microcytes appear in the RBC population because it measures the lower part of the MCV distribution curve compared to the RDW%[53]. In our case, complete blood count, measured by the hematological analyzer, showed the RDW-SD level for untreated cells in control experiments in the range of 52.5–57.5 fL with a median of 56.1 fL. However, treated with oxidative agent RBCs demonstrated increased heterogeneity (Fig. 3c), particularly at tBuOOH concentration of 1.5 mM RDW-SD was 76.3–108.5 fL with 90.6 fL median. We registered a significant change of MCV and RDW-SD parameters at tBuOOH concentration of 0.7 mM and higher, explaining the appearance of a fraction of slow (increased in their volume) cells.

For direct investigations of changes in the morphology of tBuOOH treated RBCs, we stained them with eosin-5-maleimide (EMA). This dye predominantly binds to the band 3 membrane protein and thus describes RBCs' membrane transformation[54]. Confocal microscopy registered RBCs shape changes, morphology heterogeneity, and appearance of microdomain patterns after oxidative stress induction by tBuOOH (Fig. 4a). AFM topography investigation approved the morphology heterogeneity in treated cells (Supplementary Fig. 4)

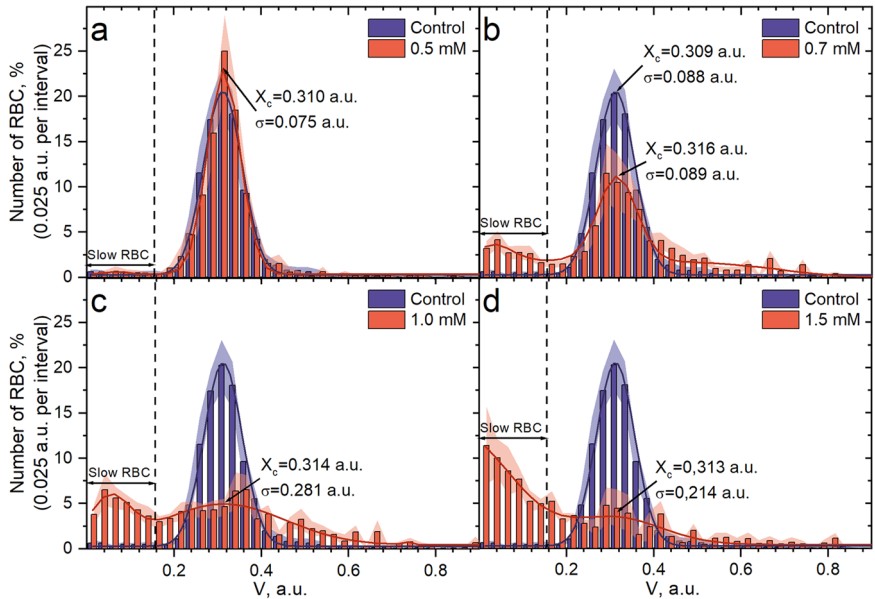

**Fig. 2 The profile of the RBCs velocity distribution in microchannels.** It was normalized to the fluid flow rate: in control, the velocity distribution is Gaussian, but at oxidant concentrations higher than 0.7 mM, it becomes bimodal: **a** 0.5 mM of tBuOOH versus control ($n = 12$ and $n = 18$ donors); **b** 0.7 mM of tBuOOH ($n = 12$ donors); **c** 1.0 mM of tBuOOH ($n = 8$ donors); **d** 1.5 mM of tBuOOH ($n = 5$ donors). The data are presented as the mean ± SE, and the error bands are shaded.

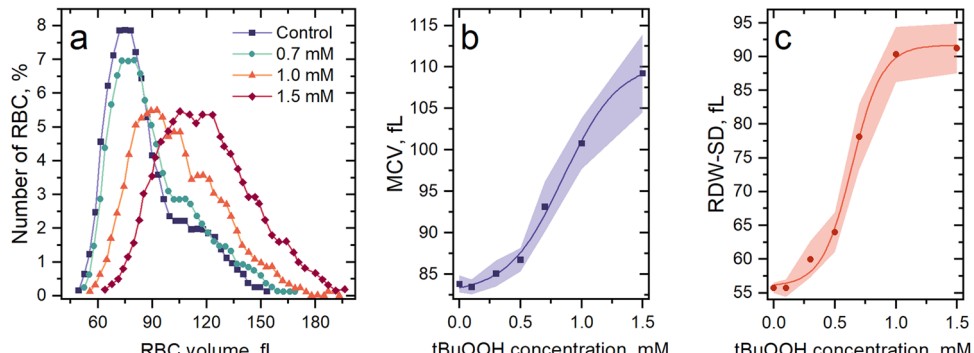

**Fig. 3 Violation of the volumetric characteristics of RBCs under the action of oxidative stress. a** Representative RBCs' RBC volume histograms show us cell swelling (digitized data of hematological analyzer). **b** MCV and **c** RDW-SD under the influence of oxidative stress, respectively. $n = 12$ donors, the data are presented as the mean ± SE values, and the error bands are shaded.

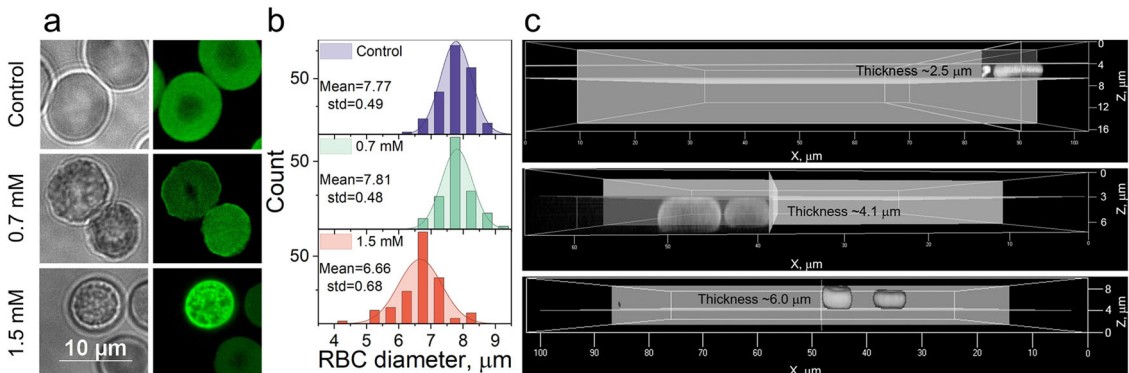

**Fig. 4 RBCs shape and morphology change under the oxidative stress, recorded by confocal fluorescence microscopy. a** Fluorescent confocal microscopy images show changes in RBC morphology under oxidative stress. **b** Histograms of RBCs diameters, calculated by confocal microscopy images. Under severe oxidative stress conditions, the diameter of RBC was lowered. **c** Z-stack fluorescent confocal microscopy images demonstrated direct dependence of RBC thickness on tBuOOH concentration. From top to bottom: 0 mM, 0.7 mM, and 1.5 mM of tBuOOH.

Additionally, the diameter of cells treated with 1.5 mM tBuOOH dropped by 16.7% (Fig. 4b); therefore, we suggested increasing their thickness. Z-stack measurements confirmed that RBCs thickness increased from ~2–3 µm in control cells to ~4–6 µm in treated with 0.7–1.5 mM of tBuOOH RBCs (Fig. 4c). This result is consistent with data obtained by the hematology analyzer, showing cell swelling, and describes the morphology of "slow RBCs", discovered by transport analysis in microfluidic channels.

Following the change of RBCs shape, we decided to analyze cell membrane transformation by the flow cytometry EMA test[54]. Most often, this test is used in clinical practice to detect hereditary spherocytosis. RBCs from patients with hereditary spherocytosis have band 3 deficiency, and therefore the EMA fluorescence signal is lower than in EMA stained RBCs from healthy people. However, our data show the appearance of an EMA brighter or EMA-positive subpopulation of cells after their treatment with tBuOOH in concentrations more than 0.7 mM (Supplementary Fig. 4), which can be associated with exposing new EMA binding sites under oxidative stress conditions. The fraction of EMA-positive cells was very close to the fraction of slow RBCs.

To be more detailed in the RBCs transformation, we analyzed the data from the flow cytometer in forward scattering and side scattering coordinates (FSC and SSC, respectively; Supplementary Fig. 5, $n = 12$ donors). Under the action of 0.7 mM, 1.0 mM, and 1.5 mM tBuOOH, we registered the accumulation of a subpopulation with a lower forward scatter, but higher side scatter signals indicated cell size decreasing and granularity increasing. Therefore, it confirmed the emergence of the subpopulation of swollen cells with the lower diameter and higher thickness.

**Direct measurements of RBCs mechanical properties**. The velocity of RBCs moving in the microchannels depends not only on their shape and volume but also on their deformation characteristics[4,5,55]. Therefore, using AFM force spectroscopy in liquid, we directly explored the changes in the mechanical properties of RBCs under oxidative stress. Obtained results showed a linear increase in Young's modulus of RBCs and their adhesion to the probe (Fig. 5) with an increase in the concentration of tBuOOH. Young's modulus of the cells almost doubled from 66.8 kPa in control samples to 131.9 kPa in samples treated with 1.5 mM of tBuOOH (Fig. 5b), whereas the adhesion force rose from 30.4 pN to 48.5 pN (Fig. 5c). Both parameters

strongly indicated a significant decrease in RBCs deformability, which along with their swelling, led to a decrease in their velocity in the microchannels. Moreover, the higher the concentration of tBuOOH was, the wider the distribution of Young's modulus was (Fig. 5a). Thus, in the RBCs population, heterogeneity increases not only in volumetric but also in deformation characteristics. To confirm it we measured cells topography in air and found that even after treating with 1.5 mM of tBuOOH there were normal cells with biconcave disc shape and damaged cells with spherical shape (Supplementary Fig. 6).

**RBCs viability under oxidative stress**. To describe the metabolic activity of the RBCs, we used the Calcein-AM (acetoxymethyl ester of Calcein) test[56]. Cells with inhibited cytoplasmic esterases are incapable of hydrolysis of nonfluorescent Calcein-AM and transform it to Calcein with bright green fluorescence. Therefore, intracellular esterase activity was calculated as mean fluorescence intensity of sample normalized to mean fluorescence intensity of control one.

Our results show that oxidative stress, caused by increased concentration of tBuOOH at the constant RBCs concentration, induced a significant decrease in Calcein fluorescence intensity (Fig. 6a, $n = 7$ donors). Esterase activity decreased exponentially in all the cells, which reflected a complex disruption of the activity of cellular supporting enzymes and a decrease in cell viability (Fig. 6b).

Another test that describes the viability of the erythrocytes is the Annexin-V test[57]. Annexin-V is a protein that binds to phosphatidylserine, which usually presents only on the inner layer of the cellular membrane. Annexin-V binding can happen in two cases: when phosphatidylserine is externalized to the outer layer of the cell membrane or when pores in the cell membrane are formed. Both events show completely nonviable cells. As the concentration of tBuOOH increased, the subpopulation of nonviable Annexin-V positive cells increased (Fig. 6b and c, $n = 7$ donors). Since the phosphatidylserine has an affinity to the spectrin cytoskeleton of RBCs[58], its externalization indicates a violation of the link between the cell membrane and the cytoskeleton. Therefore, the Annexin V test confirmed the RBCs shape and volume changes caused by oxidative stress that directly influenced the microcirculation.

Another mechanism that might be involved in RBCs shape, volume, and functional disturbance is the transformation of

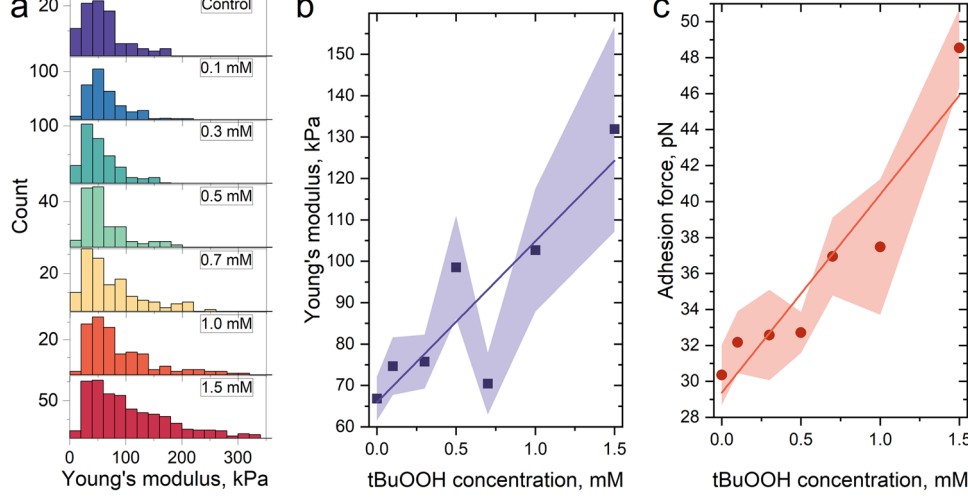

**Fig. 5 Mechanical properties of RBCs, measured by AFM. a** Representative histograms of Young's modulus distribution of RBCs, obtained from one donor; **b** Young's modulus and **c** probe-sample adhesion force after treatment with different concentrations of tBuOOH. The data are presented as the mean ± SE values, and the error bands are shaded. $n = 7$ donors (control, tBuOOH 0.1–0.5 mM), $n = 3$ donors (tBuOOH 0.7–1.5 mM).

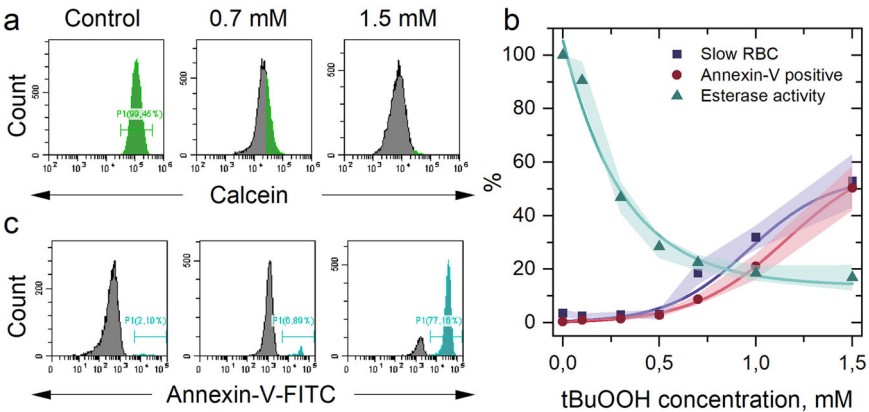

**Fig. 6 RBCs cytological characterization under oxidative stress. a** A typical histogram of flow cytometry calcein-AM test, obtained after 0.5 h of RBC incubation with tBuOOH, shows decreased esterase activity in the cells. **b** Graph of the dependence of esterase activity according to Calcein-AM test, Annexin-V positive cells fraction, and slow RBC fraction from Fig. 2 depending on the tBuOOH concentration. The data are presented as the mean ± SE values, and the error bands are shaded, $n = 7$ donors in flow cytometry tests and $n = 5$–18 donors for microfluidic tests. **c** Typical histogram of flow cytometry Annexin-V test. Cyan gate contains the Annexin-positive cells.

hemoglobin to methaemoglobin that occurred under the tBuOOH-induced oxidative stress at concentrations 0.5 mM and higher (Supplementary Fig. 7, $n = 7$ donors). Methaemoglobin cannot carry oxygen, so the RBCs functionality decreased. Besides, it competes with ankyrin for binding sites of the band 3 transmembrane protein[51]. Since ankyrin is an intermediate link between band 3 and the spectrin cytoskeleton of RBCs, the emerging competition for the binding sites leads to disruption of the connection between the band 3 protein and spectrin cytoskeleton[51]. It also leads to violations of RBCs shape and volume and impairs their transport in microchannels.

## Discussion

Erythrocytes spend about 120 days in circulation, so their population consists of cells of different ages, biochemical, and biophysical phenotypes. Therefore, oxidative stress should not affect them equally. Our studies showed that at low levels of oxidative stress biophysical properties of all the cells did not significantly change. However, esterase activity in all RBCs decreased regardless of the level of oxidative stress. Therefore, it might be used as the first sign of oxidative stress.

At high levels of oxidative stress, we observed splitting the RBCs population into two subpopulations with different biophysical properties. In the first "normal" one, the cells had biophysical and cytological properties, except esterase activity, similar to parameters of cells in control, which were not affected by oxidative stress. In the second one, called "slow", the cells showed low transit velocity in microchannels, which was correlated with membrane transformation according to EMA and Annexin-V flow cytometry tests. Other tests, such as hematological analysis, confocal microscopy, and AFM, showed that oxidative stress caused cells' swelling and increased stiffness and adhesion. Although these methods did not show cells dividing into two subpopulations due to sample preparation, measurements procedures, and low statistics, they showed exactly the same trend in changes of biophysical parameters of RBCs under oxidative stress and indicated their damage. When the concentration of tBuOOH was 0.5 mM and less all investigated biophysical and cytological parameters, averaged by all the measured cells, were close to the values in the control. The concentration of tBuOOH 0.7 mM was the critical point, after which they started changing. The only exception we found was esterase activity, which started decreasing even at concentrations of tBuOOH less than 0.7 mM. Therefore, according to our

data, we can say that RBCs at all investigated levels of oxidative stress can be in two states: "normal" and "slow" or "damaged". The switch between these two states occurred relatively fast because we did not register any intermediate states or smooth shifts in biophysical parameters at different values of oxidative stress.

The reason for such behavior of RBCs can be their unique antioxidant defense system that prevents cells from damaging during oxygen transport from lungs to tissues. It is a self-sustaining system consisting of small molecules such as vitamins E and C, glutathione[59,60], and antioxidative defense enzymes, such as superoxide dismutase[61], catalase[62,63], glutathione peroxidase, glutathione reductase, and glutathioneS-transferases[62]. Since mature erythrocytes are not capable of expressing proteins, their antioxidant defense system has a limit of the concentration of the stress inducer or its intake rate, after which it cannot cope with the stress[16,19,45,64]. Due to the different ages and cell individuality, this limit may differ for each erythrocyte. According to our data, if this system is not overloaded, the cell structure and functionality are not affected regardless of oxidative stress level. However, when the system fails, the cell very quickly loses the ability to support its shape and structure, resulting in the loss of its functionality.

Since oxidative stress increases the heterogeneity of the RBCs population, the hematological parameter RDW might be preferable for detection and analysis of damaging levels of oxidizing agents and deterioration of microcirculation than MCV. It might be useful for estimating the possible side effects of cancer chemotherapy and predicting the risk of anemia.

Interpretation of the obtained results should be performed keeping in mind several limitations, which might be answered in further studies. In our research, we induced oxidative stress adding defined amount of tBuOOH to cell suspension with constant concentration in all the experiments. Therefore, other oxidizing agents might influence the RBCs differently. Moreover, if the introduction of the oxidizing agent was significantly slower and comparable with glycolysis rate, cells' antioxidant defense system might be able to protect them from damaging by tBuOOH[19]. Another limitation is the need to attach RBCs to a surface for performing AFM and confocal microscopy in liquid media. RBCs normally exist in a suspension, therefore their immobilization efficiency and stability on the surface might be different for damaged and normal cells. This might be one of the reasons why we didn't see cells' heterogeneity on AFM and

confocal microscopy data obtained in liquid medium but saw it in AFM data obtained in air medium.

## Methods

**Blood collection**. Blood was collected by venipuncture in S-monovette tubes (Sarstedt, Nümbrecht, Germany) from 18 healthy volunteers, both sexes, the age median – 32.5 (22–68). After first centrifugation, plasma and buffy-coat layer were removed, and RBCs were washed twice by centrifugation at 400 $g$ for 3 min (Centrifuge ELMI-50CM, Riga, Latvia) and resuspended in HEPES-buffer. The blood parameters including red blood cell count (MCV) and RDW were controlled by the hematological counter Medonic-M20 (Boule Medical A.B., Sweden). The Ethics Committee of the Sechenov Institute of Evolutionary Physiology and Biochemistry RAS (Study No. 2-02; 26.02.2021) approved studies using human RBCs. All donors signed the corresponding "Informed Consent" for their participation.

**Blood preparation**. Primary buffer constituents were purchased from local Russian companies. The isotonic HEPES-buffer has the following composition, in mM: NaCl, 140; KCl, 5; HEPES, 10; MgCl2, 2; $D$-glucose, 5 EGTA 2; its osmolality 300 mOsm/kg $H_2O$ was controlled by cryoscopic osmometer Osmomat 3000 (Gonotec, Germany), pH 7.4.

Oxidative stress was induced by tBuOOH (0.1, 0.3, 0.5, 0.7, 1, and 1.5 mM; from Sigma–Aldrich, Munich, Germany), during 4–5 h, 37 °C on Thermoshaker (Eppendorf, Hamburg, Germany). Our previous study showed that it is crucial to fixate the ratio [tBuOOH]/[RBC] to acquire correct results of oxidative stress effects on RBCs[15]. Therefore, in all experiments, the concentration of erythrocytes in incubation suspension was kept constant at $0.5 \times 10^9$ cells/mL.

Validation of the oxidative stress model was made by staining RBCs with the DCF-DA dye (35845, Sigma-Aldrich), which is sensitive to the presence of ROS. To investigate their formation over time, we incubated a suspension of RBCs ($0.1 \times 10^9$ cells/mL) with 2 µM DCF-DA for 30 min at 37 °C. Then we added 0.1 mM of tBuOOH, which corresponded to our standard concentration ratio of tBuOOH 0.5 mM/RBC $0.5 \times 10^9$ cells/mL. tBuOOH was added directly to the cytometer tube. For standard flow cytometry analysis, we stained RBCs in concentration $0.5 \times 10^9$ cells/mL with DCF-DA (5 µM, 30 min, 37 °C). Then we added tBuOOH at the indicated concentrations. Flow cytometer analysis was performed 5 min after tBuOOH addition.

Hemoglobin forms were defined by measuring absorbance spectra (SPECS SSP-715-M, Spectroscopic systems, LTD, Moscow, Russia) in the range of 300–700 nm in the lysates of erythrocytes. The percentage of hemoglobin forms was calculated from the optical density of characteristic peaks at 560, 577, 630, and 700 nm using millimolar extinction coefficients[65].

**Microfluidic device fabrication**. Microfluidic devices were manufactured using soft lithography technology[47,48]. The mold was made by plasma-chemical etching of silicon to a depth of 8 µm. Etching was performed using a 100 nm thick chromium mask formed by lift-off lithography using a DWL 66 fs direct laser writing setup (Heidelberg Instruments, Heidelberg, Germany). The dimensions of the microchannels on the mold were measured by scanning electron microscope Supra 25 (Carl Zeiss, Germany) and by a high-resolution stylus-type profilometer XP-1 (Ambios Technology, Santa Cruz, California, USA).

Polydimethylsiloxane (PDMS) replicas were obtained by curing a degassed mixture of Sylgard 184 Silicone Elastomer Base and the Curing Agent 10:1 (Dow Corning, Midland, USA) at 65 °C, 4 h. After separating the PDMS replica from the mold, inlet and outlet holes were cut out using a 1 mm biopsy puncher. Then the PDMS replica was treated by oxygen plasma on a Plasma System V-15G (PINK GmbH Thermosystem, Wertheim, Germany) and covalently bonded with a $75 \times 25 \times 1$ mm glass slide. Tygon tubing was used to introduce cells into the device.

**Microfluidic experimental procedure and image analysis**. Before each experiment, microchips were filled with HEPES-buffer, which prevented the bubbles formation and the cells adhesion to the channels' walls. To ensure that only one cell passed through a microchannel simultaneously, we diluted all the samples to the concentration of $0.5 \times 10^8$ cells/mL and introduced them into the microchip under constant hydrostatic pressure.

The RBC movement in microchannels was recorded on a video camera XIMEA MC023MG-SY (XIMEA Corp., Lakewood, California, USA) with a frame rate of at least 400 frames/s, through a Leica DM4000B LED (Leica Microsystems GmbH, Wetzlar, Germany) microscope, with an N PLAN L 20×/0.40 objective (Leica Microsystems, Wetzlar, Germany). The recording was carried out in 9–10 different channels of a 16-channel microchip to obtain statistically correct data. The number of recorded RBCs during one experiment was 250–700 in control and 150–500 under the action of tBuOOH to statistics accumulation.

Image analysis was performed by a custom MATLAB (The MathWorks) script that fixed the entry (t1) and exit (t2) moments (Fig. 1b) of a single cell and calculated RBCs transit velocity. The obtained value of the velocity was normalized to the average velocity of fluid flow in a microchannel, which was defined by tracking the cell in wide channels located before and after the microchannels. Such normalization allowed comparing data from different experiments when the fluid velocity might be slightly different. For each experiment, the probability density functions were constructed and then averaged over all donors. All obtained histograms were analyzed in OriginPro 2021b (OriginLab Corporation). The "Fit multiple peaks" option determined the peak positions and their variances.

**Flow cytometry analysis**. Annexin-V-FITC test for the externalization of phosphatidylserine to the outer side of the lipid bilayer and the EMA test, which shows the transformation of band 3 transmembrane protein, were performed as following: after the incubation of RBCs with tBuOOH for 4–5 h, the aliquots RBC $5 \times 10^6$ cells/mL were stained with Annexin-V (0.1 µg/mL, 15 min, 25 °C; Biolegend, Amsterdam, The Netherlands) in HEPES-buffer with 2 mM $Ca^{2+}$; EMA (0.07 mM, 40 min, 25 °C; Molecular Probes, Eugene, USA) in HEPES-buffer. After the staining procedures, the samples were counted at once.

The activity of intracellular esterases was investigated using the Calcein-AM (56496, Sigma-Aldrich) test after 30 min incubation of RBCs with tBuOOH. This staining was performed in the presence of 2 µM Calcein-AM (Molecular Probes, Eugene, USA) for 30 min at 37 °C in 300 µL of HEPES-buffer. RBCs concentration of $5 \times 10^6$ cells/mL was used.

All the experiments were performed on flow cytometers CytoFLEX (Beckman Coulter, Brea, USA; IEPB CFC instrument) and FACSCanto (BD Biosciences, San José, USA; Alferov University's instrument) with analysis of 20,000 events. The fluorescence intensity was registered in FL1 = FITC for all fluorogenic agents used. Stained cells were not washed to minimize non-target damage. All the data were analyzed by the original software for CytoFLEX and FACSCanto.

**AFM measurements**. Young's modulus of RBCs and the probe-sample adhesion force were investigated in a liquid medium (HEPES-buffer) using a Bioscope Catalyst atomic force microscope (Bruker, Billerica, USA) by the force curves method[66]. For this experiment, DNP-10D silicon probes (Bruker, Billerica, USA) with an elastic modulus of 0.03–0.12 N/m, a resonance frequency of 12–24 kHz, and a nominal curvature radius of 20 nm were chosen. The elastic modulus of each probe was calibrated before each experiment using the thermal noise method. Before the study, the cells were attached to glass slides pretreated with a 10 µg/ml solution of poly-L-lysine for their reliable fixation. 1–4 force curve maps for each sample were constructed, $10 \times 10$ µm in size; each map contained 1–4 RBCs (Supplementary Fig. 8). To obtain force curves of the RBCs only (Supplementary Fig. 9) the regions of interests were selected manually on the maps of height. To calculate Young's modulus they were approximated by the AtomicJ program[67] using the Hertz conical model. We excluded from the dataset those values, for which the force curves had a coefficient of determination $R^2$ less than 0.9. In total, for each concentration of tBuOOH, erythrocyte samples from 3 to 7 donors and from 13 to 51 cells were analyzed.

The topography of RBCs under oxidative stress was investigated in air using SNL-C probe (Bruker, Billerica, USA) with a nominal curvature radius of 2 nm on poly-L-lysine coated glass slides.

**Confocal microscopy**. A computerized confocal laser microscope (Zeiss Axio Observer Z1 with Yokogawa spinning disk, Zeiss Microscopy, Jena, Germany), using A-Plan 100×/1.25 objective with oil immersion (Zeiss Microscopy, Jena, Germany), was used to visualize the RBCs exposed to tBuOOH. The treated RBCs were stained with EMA (20 µL of RBC with concentration $0.5 \times 10^9$ cells/mL were added to 80 µL of HEPES-buffer with 0.1 µL of 5 µg/ml EMA solution and incubated for 30 min). Then the unbound dye was removed (3000 rpm, 3 min, tabletop Elmi centrifuge), and the RBC precipitate was resuspended in 100 µL of HEPES-buffer. 10 µl of stained RBC were placed on a microscope glass slide, coated by bovine serum albumin to prevent echinocytosis.

**Statistics and reproducibility**. All the measurements were taken from distinct samples from 18 healthy donors. Each experiment was made with samples from at least three donors (Table 1). The data are presented as the mean ± SE, and the error bands on all graphs and histograms are shaded. The variables conformed to a normal distribution (Kolmogorov-Smirnov test, $p > 0.05$). The differences between the effects of tBOOH concentrations were assessed by paired $t$ test, two-tailed. The differences with $p \leq 0.05$ were considered statistically significant.

Statistical analysis was carried out using OriginPro 2021b (OriginLab Corporation), Microsoft Excel 2016 (Microsoft Corporation), and SPSS vers. 23 (IBM SPSS Statistic). Flow cytometry data were analyzed by original software CytExpert (CytoFLEX, BC) and BD FACSDiva 9 (BD Biosciences). Confocal microscopy data were analyzed by original software AxioVision Rel. 4.8 (Carl Zeiss). Hemoglobin spectrophotometry data were recorded using a spectrophotometer program SPECS SSP-715-M (Spectroscopic systems LTD, Moscow, Russia). For AFM studies was used original software Nanoscope 8.15 (Bruker) and the AtomicJ program[67].

**Table 1 Detailed statistics of all the experiments.**

| tBuOOH, mM | Number of donors | | | | | | | | | | Number of cells |
|---|---|---|---|---|---|---|---|---|---|---|---|
| | Microfluidic experiments | FSC/SSC test | EMA test | Annexin-V test | Calcein-AM test | MCV | RDW | DCF-DA | Hb forms | AFM | Confocal |
| 0 | 18 | 18 | 7 | 7 | 7 | 18 | 18 | 5 | 7 | 7 | 204 |
| 0.1 | 12 | 12 | 7 | 7 | 7 | 12 | 12 | 5 | 7 | 7 | |
| 0.3 | 12 | 12 | 7 | 7 | 7 | 12 | 12 | 5 | 7 | 7 | |
| 0.5 | 12 | 12 | 7 | 7 | 7 | 12 | 12 | 5 | 7 | 7 | |
| 0.7 | 12 | 12 | 7 | 7 | 7 | 12 | 12 | 5 | 7 | 3 | 142 |
| 1.0 | 8 | 12 | 7 | 7 | 7 | 12 | 12 | 5 | 7 | 3 | |
| 1.5 | 5 | 12 | 7 | 7 | 7 | 12 | 12 | 5 | 7 | 3 | 162 |

**Reporting summary**. Further information on research design is available in the Nature Research Reporting Summary linked to this article.

## Data availability

All datasets generated during and/or analyzed during the current study including flow cytometry and AFM data are available from the corresponding author on reasonable request. Source data can be found in Supplementary Data 1.

## Code availability

All custom code used in the current study is available from the repository Zenodo, doi: 10.5281/zenodo.6639046.

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

## Acknowledgements

This work was supported by the Russian Foundation for Basic Research, project no. 20-34-70111 "Stability". I.V.M. thanks The Ministry of Education and Science of the Russian Federation for financial support (project no. AAAA-A18-118012290371-3). We want to thank Dr. Kirill Belousov for his valuable contribution to the script's development; Dr. Ivan Morozov, Aleksandr Vorob'ev, and Konstantin Kotlyar for valuable contributions to the fabrication and characterization of microfluidic devices.

## Author contributions

A.S.B. and E.A.S. designed research. E.A.S. collected blood samples and analyzed them on a hematology analyzer. A.S.B. designed the microfluidic device and fabricated the silicon mold. E.A.S. and N.A.B. accomplished the microfluidic experiments. A.S.I. and A.S.B. developed a script for image analysis. N.A.B. performed AFM measurements. E.A.S., S.V.S., and N.A.B. carried out flow cytometry experiments. S.V.S. and N.A.B. obtained confocal microscopy images. I.V.M. contributed new reagents and research tools. N.A.B., A.S.B., and E.A.S. analyzed data. N.A.B., A.S.B., and E.A.S. contributed to writing the manuscript. All authors reviewed the manuscript.

## Competing interests

The authors declare no competing interests.
