## [Peer Review File · Communications Biology]

Reviewers' comments:

Reviewer #1 (Remarks to the Author):

The authors report an extensive characterisation of the morphological and mechanical changes that occur to red blood cells (RBC) upon induced oxidative stress by tert-Butyl hydroperoxide. It is of fundamental importance to understand how RBCs aged and are submitted to oxidative stress, for blood preservation as well as in chemotherapy.

The authors first employ a microfluidic approach to differentiate in liquid the capability of the cells to go through a micro-channel. Through this method, in combination with haematological assays, they differentiate two different populations of RBCs and an net increase of volume as a function of the tert-Butyl hydroperoxide. Then, the authors use AFM force spectroscopy to investigate the change in mechanical properties as a function of oxidative stress.

The approach, methods and statistical analysis of the manuscript are sound, thus the reviewer suggest accepting the manuscript after minor revision, where the reviewer suggest strengthening the reporting of the AFM data and analysis:

- The authors should show AFM maps of single RBC and similarly as done by confocal microscopy, do the authors observe heterogeneity at the morphological level?
- The authors should show force curves or nanomechanical/adhesion maps acquired on the RBCs. Then are the mechanical data and distribution acquired on single cells or on whole maps? Could the authors clarify why they do not see heterogeneity in mechanical properties as observed in bulk my microfluidic approach?
- The authors only mention "Although these methods did not show cells division into two subpopulations due to sample preparation and measurements procedures", could they clarify what it means more precisely?

Reviewer #2 (Remarks to the Author):

The research carried out by the authors and presented in this manuscript is of great interest to a wide range of readers. The material is well presented and easy to read.

However, I consider it necessary to make two additions to the text of the article.

1. In one or two phrases, formulate a working hypothesis of the presented research. This must be done both in the abstract and in the introduction.
2. It is necessary to add to the "Discussion" section the paragraph Limitations.

I recommend this manuscript for publication with minor revisions.

Response to the Reviewers

We would like to thank the reviewers for a positive evaluation of our manuscript and useful comments, which helped us to improve it. We made additional AFM measurements of RBCs topology under oxidative stress to strengthen the AFM data. Also we fully revised the manuscript to fix technical issues, extended the Methods and Discussion sections according to the reviewers' comments. All the changes in the manuscript were highlighted with yellow.

The point-by-point response to the reviewers' comments is below.

Reviewer #1 (Remarks to the Author):

Comment 1

The authors should show AFM maps of single RBC and similarly as done by confocal microscopy, do the authors observe heterogeneity at the morphological level?

Answer

Originally RBCs are suspension cells and in physiological conditions they exist in a liquid flow. For AFM measurements we immobilized them on glass slides with poly-L-lysine coating but didn't use any fixation chemicals such as glutaraldehyde to obtain undistorted values of Young modulus. However, the cells attached to the surface change their morphology and lyse, so we tried to make the measurements as quickly as possible. Therefore, we made them in liquid using only the Force Volume mode and didn't obtain cells topology.

To figure out if there is RBCs' heterogeneity at the morphological level we made additional AFM measurements of RBCs in air. Using obtained topological images we found that even at the concentration of 1.5 mM of tBuOOH there were cells with normal biconcave disc shape and with spherical shape. This data complement the data from confocal microscope and additionally prove the heterogeneity of RBCs population under oxidative stress. We present new AFM images in supplementary Fig. S6 and add the information about these new measurements into the main text into the Results and Methods section.

Comment 2

The authors should show force curves or nanomechanical/adhesion maps acquired on the RBCs. Then are the mechanical data and distribution acquired on single cells or on whole maps? Could the authors clarify why they do not see heterogeneity in mechanical properties as observed in bulk my microfluidic approach?

Answer

We include the Force Volume AFM maps into the Supplementary fig. S8 and the force curves into the fig. S9. We add the links to these figures into the main text of the article.

We also add in the Methods section an AFM data selection strategy. Briefly, on the 10×10 μm map, we selected RBC as the region of interest (ROI) on the map of height. The same ROI was on Young's modulus map. From the obtained data set we excluded those values, for which the force curves coefficient of determination was less than 0.9. The data presented in Fig. 5a demonstrates Young's modulus histograms for one donor (3-7 cells for each histogram). The graphs in Fig. 5b, c visualize the mean±SE of values of all the cells we measured (13-51 cells).

In the AFM experiments we didn't see statistically significant cells heterogeneity due to two possible reasons. The first one is small statistics (c.a. 1000 times smaller than in the microfluidic experiments). The second one is the immobilization efficiency and cells stability on the surface, which might be different for damaged and normal cells. However, we saw heterogeneity of the AFM images obtained in air where cells are dried and don't change their shape or detach from the surface. Therefore, scanning time is not limited which allowed us to obtain cells topography, but mechanical properties cannot be properly determined. We include this discussion into the paragraph "Limitations" in the Discussion section.

Comment 3

The authors only mention "Although these methods did not show cells division into two subpopulations due to sample preparation and measurements procedures", could they clarify what it means more precisely?

Answer

After reanalyzing all the data and making new AFM measurements we can say that confocal and AFM images show heterogeneity of cell morphology, however they don't show heterogeneity of cells' dimensions and mechanical properties. Also hematological analyzer doesn't show cell heterogeneity but indirectly it can be estimated by increasing RDW. In case of AFM studies we suppose this is due to relatively low statistics, different immobilization efficiency and stability of cells on a surface. Hematological analyzer gives the whole volume of cells determined indirectly by measuring medium resistivity when a cell moves through an aperture. We modified this phrase and moved its part into the limitations paragraph. Also we put into the Discussion section into the Limitations paragraph additional discussion about not observing cells heterogeneity by several used methods.

Reviewer #2:

Comment 1

In one or two phrases, formulate a working hypothesis of the presented research. This must be done both in the abstract and in the introduction.

Answer

According to previous in vitro experiments it is known that oxidative stress causes changes in mechanical and biophysical properties of RBCs. Also it is well known that cancer chemotherapy and several pathologies such as malaria, sepsis and diabetes cause low hemoglobin index, which causes anemia. Therefore, a working hypothesis of our research was that oxidative stress would make RBCs more rigid, which would lead to decrease of their transit velocity in microfluidic channels. We added this phrase into the abstract and introduction section.

Comment 2

It is necessary to add to the "Discussion" section the paragraph Limitations.

Answer

We added the limitations paragraph into the discussion section, which are the following:

1. We induced oxidative stress only by tert-Butyl hydroperoxide (tBuOOH). Other oxidizing agents might influence on the cells differently. Currently, we are investigating how different anticancer drugs influence on RBCs transport in the same microfluidic channels and observe similar behavior.
2. We induced oxidative stress adding defined amount of tBuOOH to cell suspension keeping its concentration constant in all the experiments. This is similar to bolus drug administration during cancer chemotherapy. If the introduction of the oxidizing agent was significantly slower and comparable with glycolysis rate, cells' antioxidant defence system might be able to protect them from damaging by oxidative stress [19].
3. AFM measurements were performed in liquid on cells, attached to the surface. RBCs normally exist in a suspension, therefore their immobilization efficiency and stability on the surface might be different for damaged and normal cells. This might be one of the reasons why we didn't see cells' heterogeneity on AFM data. To make it clear we made additional AFM measurements of cells topology in air and found that normal and damaged cells on the surface after treating them with 1.5 mM of tBuOOH. We added this data in the Supplementary Fig. 6.